# Longitudinal Metabolomics Data Analysis Informed by Mechanistic Models

**DOI:** 10.3390/metabo15010002

**Published:** 2024-12-24

**Authors:** Lu Li, Huub Hoefsloot, Barbara M. Bakker, David Horner, Morten A. Rasmussen, Age K. Smilde, Evrim Acar

**Affiliations:** 1School of Mathematics (Zhuhai), Sun Yat-sen University, Zhuhai 519000, China; 2Department of Data Science and Knowledge Discovery, Simula Metropolitan Center for Digital Engineering, 0130 Oslo, Norway; 3Swammerdam Institute for Life Sciences, University of Amsterdam, 1090 GE Amsterdam, The Netherlands; 4Laboratory of Pediatrics, Systems Medicine of Metabolism and Signaling, University of Groningen, University Medical Center Groningen, 9700 AD Groningen, The Netherlands; 5Copenhagen Prospective Studies on Asthma in Childhood (COPSAC), Herlev and Gentofte Hospital, DK-2820 Gentofte, Denmark; 6Department of Food Science, University of Copenhagen, DK-1958 Frederiksberg, Denmark

**Keywords:** challenge tests, metabolic model, (coupled) tensor factorizations, longitudinal metabolomics data, knowledge-guided machine learning

## Abstract

**Background**: Metabolomics measurements are noisy, often characterized by a small sample size and missing entries. While data-driven methods have shown promise in terms of analyzing metabolomics data, e.g., revealing biomarkers of various phenotypes, metabolomics data analysis can significantly benefit from incorporating prior information about metabolic mechanisms. This paper introduces a novel data analysis approach to incorporate mechanistic models in metabolomics data analysis. **Methods**: We arranged time-resolved metabolomics measurements of plasma samples collected during a meal challenge test from the COPSAC_2000_ cohort as a third-order tensor: *subjects* by *metabolites* by *time samples*. Simulated challenge test data generated using a human whole-body metabolic model were also arranged as a third-order tensor: *virtual subjects* by *metabolites* by *time samples*. Real and simulated data sets were coupled in the *metabolites* mode and jointly analyzed using coupled tensor factorizations to reveal the underlying patterns. **Results**: Our experiments demonstrated that the joint analysis of simulated and real data had better performance in terms of pattern discovery, achieving higher correlations with a BMI (body mass index)-related phenotype compared to the analysis of only real data in males, while in females, the performance was comparable. We also demonstrated the advantages of such a joint analysis approach in the presence of incomplete measurements and its limitations in the presence of wrong prior information. **Conclusions**: The joint analysis of real measurements and simulated data (generated using a mechanistic model) through coupled tensor factorizations guides real data analysis with prior information encapsulated in mechanistic models and reveals interpretable patterns.

## 1. Introduction

Human metabolism is a complex system, and deciphering this complex system is crucial in terms of understanding human health, diseases, and various phenotypes [1]. Metabolomics measurements of biological samples such as blood are rich sources of information, providing means to discover markers of various phenotypes, life-style differences (e.g., diet, exercise), diseases, and reveal insights about the underlying metabolic mechanisms [1,2]. Extensive biochemical knowledge including metabolic reactions is already available and has been compiled to construct computational models of human metabolism, e.g., Recon [3,4]. These models have paved the way for whole-body models (WBM) constructed based on reactions driving the underlying molecular processes, the human anatomy and the physiology [5,6]. However, there is still much to be unraveled to improve our understanding of the metabolism and to achieve precision health [7].

A step towards deciphering this complex system has been to record the functioning of the metabolism using longitudinal measurements collected over time. For instance, at short time scales, time-resolved (dynamic) metabolomics data sets collected during meal challenge tests have been used to study the human metabolic response, linking observed differences to cardiometabolic diseases [8] and various phenotypes [9]. At longer time scales, metabolomics data, e.g., collected every few months, have shown promise in terms of revealing early signs of diseases and the transition from healthy to diseased states [1,10].

The analysis of such longitudinal metabolomics data is a challenging task due to high-dimensional, noisy, and scarce/few measurements. Traditional methods rely on data summaries (e.g., data averaged across subjects [11], features summarizing time profiles using the area under the curve [12,13]) or the analysis of one feature at a time [14]. The workhorse data analysis methods in metabolomics remain univariate or multivariate methods [15] with analysis of variance [12,16] and linear mixed models [17] commonly used in longitudinal metabolomics data analysis. Rather than relying on limited views of the data such as summary statistics or one feature at a time, recent studies have arranged time-resolved metabolomics measurements as a third-order tensor (also referred to as a three-way array) with modes such as *subjects*, *metabolites*, *time* and used tensor factorizations to reveal the underlying patterns, i.e., subject groups, metabolite clusters, and temporal profiles [9,18,19,20].

Recently, there have been significant efforts under various names (e.g., informed machine learning [21], physics-informed neural networks (PINNs) [22], knowledge-guided machine learning (KGML) [23]) to incorporate prior information. These efforts mainly focus on supervised machine learning, and prior information, for instance, in the form of simulations and knowledge graphs, is integrated at the training data stage through data augmentation or used to constrain deep neural networks by simplifying architectures or penalizing loss functions [21]. In metabolomics, there are also rich sources of prior information such as computational metabolic models and knowledge bases that contain curated knowledge such as large pathway databases [24,25]. Incorporating such prior information in the analysis of metabolomics measurements holds the promise to enhance knowledge discovery by guiding the analysis of (noisy and scarce) real measurements with clean prior information. However, so far, longitudinal metabolomics data analysis approaches have been limited to data-driven methods.

In this paper, we address the question of how to incorporate the prior information encapsulated in computational metabolic models in metabolomics data analysis. We address this question in the context of unsupervised learning in order to develop methods that can analyze longitudinal metabolomics measurements and reveal unknown subject stratifications to facilitate precision health. We introduce a novel data fusion approach that jointly analyzes simulated data generated using a mechanistic model and real data. In particular, we focus on the analysis of time-resolved metabolomics measurements of plasma samples collected during a meal challenge test. We arrange the real data as a third-order tensor: *subjects*, *metabolites*, *time samples*. Simulated challenge test data generated using a human whole-body metabolic model are also arranged as a third-order tensor: *virtual subjects*, *metabolites*, *time samples*. Real and simulated data sets, which are coupled in the *metabolites* mode, are then jointly analyzed using coupled tensor factorizations (see Figure 1). We demonstrate that by guiding the analysis of noisy real data using clean simulated data, the proposed joint analysis approach achieves improved performance in terms of pattern discovery, revealing patterns with higher correlations with a BMI (body mass index)-related phenotype compared to the analysis of only real data. We also demonstrate the advantages of incorporating prior information using such a data fusion approach in the presence of incomplete measurements and discuss the limitations in the presence of wrong prior information.

## 2. Materials and Methods

### 2.1. Real Meal Challenge Test Data

The real data corresponded to measurements of specific hormones and Nuclear Magnetic Resonance (NMR) spectroscopy measurements of blood samples collected during a meal challenge test from the COPSAC_2000_ cohort [26]. The cohort consisted of 411 healthy subjects (with mothers with a history of asthma). The data in this paper came from 299 of those generally healthy subjects who underwent a meal challenge test at the age of 18. Blood samples were collected from the participants after overnight fasting and also following a standardized mixed meal [27]. The meal was a hot beverage consisting of palm oil, glucose, and skimmed milk powder. Blood samples were collected at 15, 30, 60, 90, 120, 150, and 240 min after the meal intake. The COPSAC_2000_ study was conducted in accordance with the Declaration of Helsinki and was approved by the Copenhagen Ethics Committee (KF 01-289/96 and H-16039498) and the Danish Data Protection Agency (2015-41-3696). The study participants gave written consent.

Plasma samples were then measured using NMR through the Nightingale Blood Biomarker Analysis, which provides 250 features for each sample. These features include lipoproteins, apolipoproteins, amino acids, fatty acids, glycolysis-related metabolites, ketone bodies, and an inflammation marker. Details about the meal challenge test, sample preparation, and the full list of features are given in [9]. For each participant, additional metainformation including body composition measures and HOMA-IR (Homeostatic model assessment for Insulin Resistance, an insulin resistance measure) was also available. Descriptive statistics of these metavariables (i.e., HOMA-IR, weight, height, waist circumference, BMI, waist/height ratio, muscle mass, fat mass, body fat percentage, muscle to fat ratio, fat mass index, and fat-free mass index) stratified by sex are given in [9].

To investigate metabolic differences among subjects in response to a meal challenge, we previously analyzed both fasting and T0-corrected data (where postprandial data were corrected by subtracting the fasting state measurements) for this cohort. Our analysis revealed static and dynamic biomarkers of a BMI-related phenotype, as well as gender-related differences [9,28]. In this study, we considered six features out of the complete set of measurements since these six features were the common blood metabolites in the WBM model, including insulin (Ins), glucose (Glc), pyruvate (Pyr), lactate (Lac), alanine (Ala), and β-hydroxybutyrate (Bhb). We focused on analyzing T0-corrected data, as previous studies have demonstrated their effectiveness in terms of capturing dynamic biomarkers [9,29]. Six subjects (three males and three females) were removed before the analysis since two subjects had a large quantity of missing data, and four subjects had extreme levels of acetate (greater than 0.4 mmol/L) probably due to a recent alcohol exposure. Measurements were arranged as a third-order tensor with the following modes: *subjects*, *metabolites*, *time samples*. The tensor was of size 141 subjects × 6 metabolites × 7 time samples for males, and 152 subjects × 6 metabolites × 7 time samples for females.

### 2.2. Simulated Meal Challenge Test Data

We generated simulated postprandial metabolomics data using a human whole-body metabolic model, which involved 202 metabolites, 217 reaction rates, and 1140 kinetic parameters [6]. The model was based on ordinary differential equations, capturing the complexity of multi-organ interactions and incorporating insulin and glucagon regulation after a meal intake. In that model, a meal containing 87 g carbohydrate and 33 g fat was considered after a 10-hour fasting. Note that the simulated meal composition differed from that of the real meal, but the responses of key glycolysis-related metabolites to the meal challenge were in the physiological range. A detailed comparison of the real and simulated meal as well as the time profiles following the meal challenge are available in [29]. For each subject, the data were generated as follows: First, 10-hour fasting concentrations were acquired for each individual by running the human WBM model with a unique set of randomly perturbed kinetic constants in the liver. We used the default initial values for model variables from [6], but specific adjustments were made for initial concentrations of certain blood metabolites, i.e., insulin, glucose, pyruvate, lactate, alanine, β-hydroxybutyrate, triglyceride, and total cholesterol were set to the median of fasting concentrations in the real data. Subsequently, the simulation advanced to the meal challenge phase and ran the human WBM model using the 10 h fasting state of each individual as initial values. Concentrations of metabolites were recorded at specific time points (aligned with the measurements in real data). The metabolic model provided simulated concentrations of 202 metabolites from the blood and eight different organs. In this study, we used the concentrations of six blood metabolites, i.e., Ins, Glc, Pyr, Lac, Ala, and Bhb, which were also measured in real data.

We generated 50 virtual control subjects without introducing any group differences, and with individual variations introduced through random perturbations of the kinetic parameters in the liver. For each kinetic parameter, a random perturbation of up to 20% of its default value was introduced. The simulated data are available on GitHub (accessed on 19 December 2024): https://github.com/Lu-source/project-of-challenge-test-data/. See [29] for more details on individual variations. We observed that a sample size of around 50 or more subjects was needed in order to extract robust patterns. With a smaller number of subjects, e.g., 10 subjects, only idiosyncratic behavior was captured. The simulated T0-corrected metabolomics data were arranged as a third-order tensor of size 50 subjects × 6 metabolites × 7 time samples. Note that since our goal was to guide the analysis of real data through clean patterns extracted from virtual subjects, and the real subjects in the cohort were healthy, when generating data for virtual subjects, we kept the individual variation low and did not introduce any patterns that would be expected in diseases.

### 2.3. Tensor Factorizations

As an extension of matrix factorizations to higher-order data sets (also known as multiway arrays), tensor factorizations are used to extract the underlying patterns in multiway arrays [30,31]. They have been successfully used in many domains including neuroscience [32,33], chemometrics [34], and social network analysis [35]. Among different tensor factorization methods, we focused on the CANDECOMP/PARAFAC (CP) tensor model [36,37], also known as the canonical polyadic decomposition [38] to analyze time-resolved metabolomics data sets. The CP model extracts the underlying patterns uniquely under mild conditions [31]. Uniqueness properties of the CP model facilitate interpretation, which is particularly important when the goal is to discover phenotypes and biomarkers in metabolomics data analysis.

Given a third-order tensor $X\in R^{I\times J\times K}$, an *R*-component CP model represents the data as the sum of minimum number of rank-one tensors as follows: $$X\approx \sum_{r=1}^{R}a_{r}\circ b_{r}\circ c_{r},$$
where ∘ denotes the vector outer product, and $a_{r}$, $b_{r}$, and $c_{r}$ correspond to the *r*th column of factor matrices $A\in R^{I\times R}$, $B\in R^{J\times R}$ and $C\in R^{K\times R}$, respectively. The CP model can also be denoted as X≈〚λ;A,B,C〛. $\lambda \in R^{R}$ can absorb the weights of the rank-one tensors, i.e., $a_{r}\circ b_{r}\circ c_{r}$ for r=1,…,R, by normalizing columns of the factor matrices to unit norm. The rank-one components reveal the underlying patterns in the data, e.g., if X is a *subjects* by *metabolites* by *time samples* tensor, the *r*th CP component may reveal subject stratifications in $a_{r}$, groups of metabolites responsible for the stratification in $b_{r}$, and their temporal pattern in $c_{r}$. The CP model is unique up to permutation and scaling ambiguities under mild conditions, where the permutation ambiguity indicates that the order of rank-one tensors is arbitrary, and the scaling ambiguity corresponds to arbitrarily scaling the vectors in each rank-one tensor as long as the product of the norms stays the same. These ambiguities do not interfere with the interpretation of the extracted patterns.

The CP model is often fit to the data by solving the following optimization problem:(1)$$\min_{A,B,C}∥X\;-\;〚A,B,C〛{∥}^{2},$$
where $∥\;\cdot \;∥$ denotes the Frobenius norm for matrices and higher-order tensors, and 2-norm for vectors. The Tikhonov regularization can be included by adding $\gamma ({∥\;A\;∥}^{2}+{∥\;B\;∥}^{2}+{∥\;C\;∥}^{2})$ to the objective function. In the presence of missing entries in X, the CP model can be fit to the data by solving the following weighted optimization problem, which fits the CP model only to the known entries in tensor X and ignores the missing entries [39]:(2)$$\min_{A,B,C}∥\;W*(X-〚A,B,C〛){∥}^{2},$$
where W is a binary tensor, i.e., $w_{ijk}=1$ if $x_{ijk}$ is known, and $w_{ijk}=0$ if $x_{ijk}$ is missing. The symbol ∗ denotes the (element-wise) Hadamard product.

As a result of its interpretability, CP model-based approaches have been widely used in many applications, e.g., longitudinal microbiome data analysis [40], the analysis of neuroimaging signals [32,41], when the goal is to interpret the underlying patterns to extract insights from complex data. Recent studies have demonstrated the promise of the CP model in terms of revealing subject stratifications/phenotypes and underlying metabolic mechanisms from time-resolved metabolomics data [18], in general, and longitudinal metabolomics measurements collected during meal challenge tests [9,20,29], in particular. Alternative tensor factorization methods such as the higher-order singular value decomposition (HOSVD) have also recently been used to analyze metabolomics measurements collected before and after an oral glucose challenge test [19]. HOSVD imposes additional constraints for uniqueness such as the orthogonality constraints on the factor matrices. As such strong constraints imposed for uniqueness may limit the interpretability of the extracted patterns, we relied on the CP model to analyze longitudinal metabolomics measurements.

### 2.4. Coupled Tensor Factorizations

In this paper, we considered incorporating the prior knowledge encapsulated in computational models of human metabolism in the analysis of longitudinal metabolomics measurements. We jointly analyzed simulated data (in the form of tensor Y of size $I_{\mathrm{virtual}}$ by $J_{\mathrm{virtual}}$ by $K_{\mathrm{virtual}}$ with *virtual subjects*, *metabolites*, and *time samples* as modes), generated using a human WBM model, and the real data (in the form of tensor X of size $I_{\mathrm{real}}$ by $J_{\mathrm{real}}$ by $K_{\mathrm{real}}$ with *subjects*, *metabolites*, and *time samples* as modes). We focused on the case where X and Y were coupled in the *metabolites* mode as in Figure 1 with $J_{\mathrm{virtual}}=J_{\mathrm{real}}=J$, i.e., only common metabolites in simulated and real data were taken into account in the analysis.

An effective approach to jointly analyze such higher-order data sets is to use coupled tensor factorizations [35,42,43,44]. Given tensors X and Y coupled in the *metabolites* mode, a coupled CP model jointly analyzes them by modeling each tensor using a CP model and extracting the same factor matrix from the coupled mode, e.g., the same B factor matrix in both CP models as in Figure 1. In particular, we used the structure-revealing CMTF model (also known as the advanced Coupled Matrix and Tensor Factorizations (ACMTF)) [45], which jointly analyzes each data set while trying to learn shared and unshared factors.

Given $X\in R^{I_{\mathrm{real}}\times J\times K_{\mathrm{real}}}$ and $Y\in R^{I_{\mathrm{virtual}}\times J\times K_{\mathrm{virtual}}}$ coupled in the second mode, e.g., *metabolites* mode, an *R*-component ACMTF model jointly analyzes these data sets by solving the following optimization problem:(3)$$\begin{array}{cc} \min_{\lambda ,\sigma ,A,B,C,D,E} & ∥X-〚\lambda ;A,B,C〛{∥}^{2}+∥\;Y-〚\sigma ;D,B,E〛{∥}^{2}+\beta {∥\;\lambda \;∥}_{1}+\beta {∥\;\sigma \;∥}_{1}\\ s.t.\;\; & ∥\;a_{r}\;∥=∥\;b_{r}\;∥=∥\;c_{r}\;∥=∥\;d_{r}\;∥=∥\;e_{r}\;∥=1,\mathrm{for}\;r=1,\cdots ,R, \end{array}$$
where columns of factor matrices $A\in R^{I_{\mathrm{real}}\times R},B\in R^{J\times R},C\in R^{K_{\mathrm{real}}\times R},D\in R^{I_{\mathrm{virtual}}\times R},E\in R^{K_{\mathrm{virtual}}\times R}$ are constrained to be unit norm, i.e., $∥\;a_{r}\;∥=1$, $∥\;b_{r}\;∥=1$, $∥\;c_{r}\;∥=1$, $∥\;d_{r}\;∥=1$, and $∥\;e_{r}\;∥=1$ for r=1,⋯,R. With normalized factor vectors in each mode, $\lambda \in R^{R}$ and $\sigma \in R^{R}$ contain the weights of the rank-one terms in each data set. ${∥\;\cdot \;∥}_{1}$ denotes the 1-norm of a vector. By enforcing sparsity on the weights through the 1-norm penalty, when β>0, the ACMTF model tries to reveal unshared factors with zero or close to zero weights. Coupled CP models inherit uniqueness properties from the CP model [44], and sparsity penalties on the weights have been effective to reveal shared and unshared factors [45]. Note that unit norm constraints already act as regularization; therefore, we did not consider further regularization of the model. As for the CP model, the ACMTF model can also be fit to incomplete data sets using a weighted optimization through the use of binary tensor(s) W as in Equation (Equation 2) [45].

Other types of coupling between real and simulated data sets are also possible. For instance, data sets can be coupled in the *time* mode. However, uncoupled time patterns may facilitate the discovery of discrepancies between simulations and real data, as we observed in the experiments. Data sets can also be coupled in both *metabolites* and *time* modes through concatenation in the *subjects* mode and analyzed as a single tensor, for instance using a CP model. In such a setting, the model would focus on modeling the difference between virtual and real subjects rather than focusing on stratifications among real subjects. Therefore, in this paper, we considered coupling the data sets only in the *metabolites* mode.

CMTF models have been used in many domains, e.g., social network analysis [35], remote sensing [46], neuroscience [41,47], and chemometrics [42]. Recently, we used a CMTF model to jointly analyze fasting state (T0) and (T0-corrected) dynamic metabolomics data coupled in the *subjects* mode to extract static and dynamic markers for the same subject stratifications [28]. Coupled factorizations have been successfully used in relational data analysis by incorporating additional side information in the form of matrices/higher-order tensors and jointly analyzing them, for instance, for missing link prediction [48], or temporal phenotyping [49], to name a few applications. However, to the best of our knowledge, this is the first application of such a model to incorporate mechanistic models in unsupervised data mining.

### 2.5. Experimental Set-Up

#### 2.5.1. Data Preprocessing

Before the analysis, third-order tensors (X and Y) representing T0-corrected data with modes *subjects* by *metabolites* by *time samples* were centered across the *subjects* mode to remove mean-based offsets and then scaled within the *metabolites* mode (i.e., each metabolite slice was divided by the root-mean-square value of that slice) to ensure similar scales across all metabolites. See [50] for the centering and scaling of higher-order tensors. After preprocessing, each tensor was divided by its Frobenius norm to give equal importance to each data set in (Equation 3).

#### 2.5.2. Implementation Details

We used acmtf_opt from the CMTF toolbox (https://github.com/eacarat/CMTF_Toolbox, accessed on 19 December 2024) to fit the ACMTF model [45], and cp_wopt [39] from the Tensor Toolbox (version 3.1) [51] to fit the CP model. The nonlinear conjugate gradient algorithm from the Poblano Toolbox [52] was used to solve the optimization problems for CP and ACMTF. In the case of missing entries, functions acmtf_opt and cp_wopt use weighted optimization fitting the model only to the known entries as in Equation (Equation 2). The sparsity penalty parameter β in (Equation 3) was set to $10^{-3}$ [45]. Multiple random initializations were used to avoid local minima when fitting ACMTF and CP models, and the initialization that yielded the lowest function value was chosen as the best run for further analysis. All experiments were conducted using MATLAB 2020a. For more details, see the GitHub repository (https://github.com/Lu-source/ACMTF_Real_Simulated, accessed on 19 December 2024).

#### 2.5.3. Model Selection

Selecting the right number of components (*R*) is crucial for extracting the underlying patterns accurately for CP and ACMTF models. Here, we relied on the replicability of the extracted patterns across random subsets of real subjects [9,53]. The replicability was assessed by fitting the model (CP or ACMTF) to subsets of subjects, where 10% of subjects from the real data X in Figure 1 were randomly removed. The similarity of the patterns extracted from different subsets of subjects (after finding the best matching permutation) was computed using the factor match score (FMS). ${\mathrm{FMS}}_{X}$ and ${\mathrm{FMS}}_{y}$ are defined as:$$\begin{array}{cc} {\mathrm{FMS}}_{X} & =\frac{1}{R}\sum_{r=1}^{R}\frac{|{b_{r}}^{T}{\hat{b}}_{r}|}{∥\;b_{r}\;∥∥\;{\hat{b}}_{r}\;∥}\frac{|{c_{r}}^{T}{\hat{c}}_{r}|}{∥\;c_{r}\;∥∥\;{\hat{c}}_{r}\;∥},\\ {\mathrm{FMS}}_{Y} & =\frac{1}{R}\sum_{r=1}^{R}\frac{|{d_{r}}^{T}{\hat{d}}_{r}|}{∥\;d_{r}\;∥∥\;{\hat{d}}_{r}\;∥}\frac{|{b_{r}}^{T}{\hat{b}}_{r}|}{∥\;b_{r}\;∥∥\;{\hat{b}}_{r}\;∥}\frac{|{e_{r}}^{T}{\hat{e}}_{r}|}{∥\;e_{r}\;∥∥\;{\hat{e}}_{r}∥}, \end{array}$$
where $\langle b_{r},c_{r},d_{r},e_{r}\rangle$ and $\langle {\hat{b}}_{r},{\hat{c}}_{r},{\hat{d}}_{r},{\hat{e}}_{r}\rangle$ are the *r*th column of factor matrices from *R*-component ACMTF models in the *metabolites* (coupled mode), *time* (real), *subjects* (virtual), and *time* (virtual) modes. When assessing the replicability of CP models, only ${\mathrm{FMS}}_{X}$ was considered. FMS values are between 0 and 1, where 1 indicates an exact match between the components of models fitted to different subsets of subjects.

The *model fit* was also computed to determine how well a model explained the data. The fit was defined as: (4)$$\begin{array}{l} {\mathrm{Fit}}_{X}(\%)=(1\;-\;\frac{{∥W\;*\;(X\;-\;\hat{X})∥}^{2}}{{∥\;W\;*\;X\;∥}^{2}})\times 100,\\ {\mathrm{Fit}}_{Y}(\%)=(1\;-\;\frac{{∥Y\;-\;\hat{Y}∥}^{2}}{{∥Y∥}^{2}})\times 100, \end{array}$$
where $\hat{X}=〚\lambda ;A,B,C〛\;\mathrm{and}\;\hat{Y}=〚\sigma ;D,B,E〛$ are the approximations of real and simulated data, respectively. The binary tensor W indicates observed $\left(w_{ijk}=1\right)$ or missing $\left(w_{ijk}=0\right)$ entries in tensor X A fit value close to 100% implies that the model explains the data well. 

## 3. Results

Previously, gender differences were observed in the COPSAC_2000_ cohort in terms of how the dynamic metabolic response was related to a BMI-related group difference [9]. In particular, while similar dynamic metabolic response patterns were observed in males vs. females vs. all subjects (males and females combined), how those patterns related to BMI groups and correlated with metavariables showed differences in males vs. females [9]. Therefore, in order to avoid having the results affected by gender differences, we analyzed males and females separately. In males, we demonstrated that the joint analysis of simulated and real metabolomics data had better performance in terms of pattern discovery, achieving higher correlations with a BMI-related phenotype compared to the analysis of only real data. In females, the joint analysis of simulated and real data sets achieved similar performance compared to the analysis of only real data. Using a larger set of metabolites from the same meal challenge study, we previously observed that correlations were much lower for females possibly due to anthropometric differences (e.g., where fat is deposited in males vs. females) [9]. Here, considering a specific set of metabolites, higher correlations were achieved in females; however, they were still much lower than in males. As the improvement may be limited due to anthropometric differences and the WBM model validation mainly relies on data from males [6], in the rest of the paper, we focus on the analysis of measurements from males and discuss the results for females in the Discussion section.

We evaluated the performance of the methods in terms of how well they revealed a BMI-related phenotype characterized by the metavariables. More specifically, the performance was assessed in terms of the correlations between subject scores (captured by the methods) and BMI-related metavariables. Note that as the subjects in the cohort were generally healthy, there was no variation due to any specific disease that was expected to be captured by the methods.

### 3.1. Analysis of Real Metabolomics Data

We analyzed the real T0-corrected metabolomics measurements from males X (141 males × 6 metabolites × 7 time points) using a three-component CP model with Tikhonov regularization (with regularization parameter γ=0.01). See Appendix A, for the selection of number of components and regularization parameter. The model explained 52.4% of the data. Figure 2 shows the factors extracted by the three-component CP model.

We observed that the model revealed a weak BMI-related group difference in the second component (*p*-value $=1\times 10^{-4}$ using a two-sample *t*-test on $a_{2}$ in Figure 2), where *Lower BMI* and *Higher BMI* correspond to BMI <25 and BMI ≥25, respectively. In the metabolites mode ($b_{2}$), Ins, Glc, and Ala had the largest score values (in terms of absolute value) indicating that they were the most related metabolites to the BMI group difference. In particular, Ins and Glc had positive values indicating that changes in these metabolites were positively related to *Higher BMI* while the change in Ala was negatively related to *Higher BMI*. The *Higher BMI* group consisted of overweight and obese individuals. Obesity, especially intra-abdominal adiposity, is known to be linked to issues in glucose metabolism and insulin resistance [54]. In insulin resistance, glucose levels go up after a meal intake and stay high which results in pancreas releasing more insulin, therefore resulting in the positive relation with insulin/glucose and the higher BMI group. The negative association between the change in Ala after meal intake and BMI groups has also been highlighted in a recent study [55]. In the time mode, $c_{2}$ increased until around 1.5 h and decreased afterwards showing the temporal profile of the metabolic response modeled by this component. Although we discuss subject group differences based on BMI groups, $a_{2}$ was also correlated with other metavariables, as shown in Figure 4a.

The first component $\langle a_{1},b_{1},c_{1}\rangle$ and the third component $\langle a_{3},b_{3},c_{3}\rangle$ in Figure 2 potentially modeled non-BMI related individual differences in the data. The first component mainly modeled an early response captured by $c_{1}$, and the third component modeled a late response captured by $c_{3}$. The third metabolite factor, i.e., $b_{3}$, revealed that changes in Pyr, Lac, and Ala behaved opposite to the change in Bhb, which aligned with the observation that the concentrations of Pyr, Lac, and Ala increased while Bhb decreased after the meal intake, as shown in the temporal profiles of these metabolites in Appendix A. No statistically significant BMI-related group difference was observed in these components, and correlations between subject scores and metavariables were less than or around 0.2 (except for HOMA-IR, for which the third component revealed a correlation of 0.32).

### 3.2. Joint Analysis of Real and Simulated Metabolomics Data

We jointly analyzed the real T0-corrected metabolomics data X (141 males × 6 metabolites × 7 time points) and simulated metabolomics data Y (50 subjects × 6 metabolites × 7 time points) using a three-component ACMTF model by coupling the data sets in the *metabolites* mode. The model explained 50.0% of the real data and 71.8% of the simulated data. See Appendix A, for the selection of the number of components.

Figure 3 shows the factors extracted using a three-component ACMTF model. The second component ($a_{2}$) revealed a BMI-related group difference (*p*-value $=3\times 10^{-6}$). In the metabolites mode $b_{2}$, Ins and Glc had the largest absolute score values showing positive association with *Higher BMI*. The first and third components may be modeling non-BMI related individual variations in the data. No statistically significant BMI-related group difference was observed in these components, and correlations with metavariables were much smaller than 0.2, with the largest ones around 0.2. These components had similar dynamic patterns in the real data part (i.e., $c_{1}$ and $c_{3}$); however, corresponding metabolite factors, $b_{1}$ and $b_{3}$, modeled the behavior of different metabolites, i.e., Pyr, Lac, and Ala had large score values on $b_{1}$ while Bhb was mainly modeled by $b_{3}$.

We did not observe any apparent clustering among the virtual subjects ($d_{1}$, $d_{2}$, and $d_{3}$) since no group-specific information was incorporated during the generation of the simulated data. Consequently, virtual subject patterns mainly reflected individual variations in the simulated data.

As the data sets were coupled only in the metabolites mode through an ACMTF model, the model revealed time profiles specific to each data set. When time profiles $c_{1}$, $c_{2}$, $c_{3}$ extracted from the real data were compared with the ones from the simulated data, i.e., $e_{1}$, $e_{2}$, $e_{3}$, we observed that the model captured different temporal profiles from real and simulated data potentially revealing the discrepancies between simulated and real data. This discrepancy may be due to the fact that virtual and real subjects underwent meal challenges with different contents, and this may have resulted in the observed differences in temporal profiles as discussed in [29].

### 3.3. Analysis of Real Data vs. Joint Analysis of Simulated and Real Data

Figure 2 and Figure 3 show that the joint analysis of simulated and real data revealed cleaner patterns in the metabolites mode, less affected by the noise in real data: (i) Pyr, Lac, and Ala were close to each other with large score values in both $b_{1}$ and $b_{3}$ in Figure 2; on the other hand, they were mainly modeled by $b_{1}$ using the joint analysis as shown in Figure 3. (ii) Ins and Glc clustered closely with large score values in both $b_{1}$ and $b_{2}$ in Figure 2 while they were modeled mainly by the second factor ($b_{2}$ in Figure 3) in the joint analysis. (iii) Bhb had contributions in all components in Figure 2, whereas in Figure 3, Bhb only contributed to $b_{3}$. As a result of these differences, we observed that the BMI-related component captured through the joint analysis of simulated and real data, i.e., $b_{2}$ in Figure 3, showed higher correlations with all metavariables (Figure 4a).

Another observation is that the CP analysis of real data revealed a potential negative association between Ala and the *Higher BMI* group (as shown in $\langle a_{2},b_{2},c_{2}\rangle$ in Figure 2). This pattern was not evident in the joint analysis. Time profiles of raw data including Ala are given in Appendix A. These plots show differences (which are statistically significant at some time points) in Ala concentrations in *Higher* vs. *Lower BMI* groups. The joint analysis did not identify Ala as an important metabolite related to BMI group difference since the simulated data did not support such a pattern. For virtual subjects, we observed different temporal profiles for Ala compared to Ins, Glc, and Bhb, and similar temporal profiles compared to Pyr and Lac. This prevented the joint analysis from extracting a metabolite pattern similar to $b_{2}$ in Figure 2 and facilitated the extraction of $b_{1}$ and $b_{2}$ in Figure 3. For the CP analysis of only simulated data, see Appendix A.

### 3.4. Joint Analysis of Real and Simulated Metabolomics Data in the Presence of Missing Data

Missing data may be observed in a metabolomics data analysis due to various reasons such as preprocessing issues of the raw metabolomics measurements or sample handling problems. These errors may cause random missing measurements or missing measurements for a whole sample. Real metabolomics measurements in our experiments, i.e., tensor X, had only 1.5% of the tensor entries missing, and that did not pose any challenges for data analysis. The analysis of tensor X using a CP model and its joint analysis with simulated data using an ACMTF model were carried out by fitting the models to the known entries in real data as discussed in Section 2, and we discussed the results in Section 3.1 and Section 3.2. Here, in order to demonstrate the performance of the joint analysis of real and simulated data in the presence of a significant (but still realistic) number of missing measurements in real data, we introduced additional missing entries in tensor X. More specifically, we randomly set 10% of the real data to be missing, including 5% of the data corresponding to missing fibers. A fiber corresponded to measurements of all metabolites from a sample, i.e., from a specific subject at a certain time point. The remaining 5% corresponded to randomly missing entries (i.e., a single measurement). The incomplete real measurements were then analyzed using a CP model and also jointly analyzed with simulated data using an ACMTF model. We generated 32 such randomly incomplete data sets. Figure 4b reports the correlations (of the subject scores from the BMI-related component) with metavariables when using an ACMTF model vs. a CP model. Boxplots correspond to the correlations from the analysis of 32 data sets. Figure 4b shows that the joint analysis demonstrated more consistent and higher correlations than the CP model.

### 3.5. Joint Analysis of Real and Simulated Metabolomics Data in the Presence of Conflicting Information

While we have demonstrated the effectiveness of the joint analysis of real and simulated data, it is important to note that it is possible to have conflicting information between the prior information (e.g., simulated data) and real data. Such conflicting information may prevent revealing the underlying patterns accurately. To demonstrate the performance of the joint analysis in the presence of conflicting information, we created simulated data with wrong prior information as follows:Step 1. *Default patterns*. We used a three-component CP model to extract the underlying patterns from the simulated T0-corrected data (see Appendix A). The data approximated by the model were denoted by $\hat{Y}$, and residuals by E.Step 2. *Conflicting pattern construction*. The first and third components (from Step 1) were retained, while the second component was modified by introducing wrong prior information. In the default (correct) pattern, Ins and Glc were close to each other, having large positive values in the second component, while values of the remaining metabolites were close to zero. We broke down the positive association between Ins and Glc and set the loading values of Ins, Glc, Pyr, Lac, Ala, and Bhb to 1, −1, 0, 0, 0, and 0, respectively (the factor vector was then normalized, i.e., divided by its two-norm). See Appendix A for the modified pattern. This is wrong prior information for the real data, which consisted of healthy subjects, and no such relation between Ins and Glc was expected.Step 3. *Construction of simulated data with conflicting information*. Tensor ${\hat{Y}}_{1}$ was then constructed using the modified CP patterns. The simulated data with conflicting information, denoted by ${\hat{Y}}_{1}$, were obtained by adding the residual term (obtained in Step 1) to ${\hat{Y}}_{1}$, i.e., $Y_{1}={\hat{Y}}_{1}+E.$


Figure 5 shows that coupling with conflicting prior information led to worse correlations between subject scores and metavariables compared to the analysis of real
data using a CP model. Such poor correlations stemmed from the fact that the joint analysis obstructed the extraction of correct patterns from the real data. This issue occurred because these correct patterns, in particular the second component in Figure 3 mainly modeling Ins and Glc, were substantially different from the “broken-down” pattern, i.e., the second component in Appendix A, in the simulated data. We observed that the ACMTF model extracted the “broken-down” pattern (see
Appendix A). Moreover, we also observed that the ACMTF model explained less of the real data and more of the simulated data compared to the case
when the real data were jointly modeled with the default simulated data. In other
words, the model fit of the real data part dropped from 50.0% to 44.0% while the model
fit of the simulated data increased from 71.8% to 74.3% when the default simulated data were replaced with the simulated data containing conflicting information in the joint analysis. When we looked at weights of the components (i.e., ***λ***, ***σ*** in Figure 1) learned by ACMTF models given in Figure 6a,b, we observed a decrease in the weight of the second component (Ins–Glc-related pattern) in the real data part (i.e., *λ*_2_) while *λ*_1_ and *λ*_3_ remained relatively unchanged, which is consistent with the observed decrease in the model fit. Although we observed a decrease in *λ*_2_, the second component still looked like a shared pattern. *λ*_2_ close to zero in that case would indicate an unshared factor and would
make the identification of conflicting information possible. This shows the limitation of the ACMTF model in terms of detecting conflicting information in the case of noisy data sets.

## 4. Discussion

In this paper, we jointly analyzed simulated data (generated using a human WBM model) and time-resolved metabolomics measurements using coupled tensor factorizations. Our experiments demonstrated that the proposed approach achieved better pattern discovery performance compared to the analysis of only real data. A similar performance improvement was also demonstrated in the presence of real data with missing entries. This enhanced performance was attributed to the extraction of cleaner patterns facilitated by the joint analysis of real data with clean simulated data.

Compared to the improved performance in males, informing the real data analysis with simulated data through the joint analysis of real and simulated metabolomics measurements did not change the performance much in females. We analyzed the real T0-corrected metabolomics measurements from females X (152 females × 6 metabolites × 7 time points) using a three-component CP model with Tikhonov regularization (with γ=0.01) and jointly analyzed real and simulated data using a three-component ACMTF model. The model fit was 50.7% for the CP model and one of the CP components captures a BMI-related group difference (*p*-value $=3\times 10^{-5}$). For the ACMTF model, fit values were 48.5% and 71.8% for real and simulated data sets, respectively. One of the ACMTF components also revealed a BMI-related group difference (*p*-value $=7\times 10^{-4}$). Figure 7 shows that correlations with metavariables obtained using a CP model were comparable to the ones obtained using the joint analysis. In both males and females, the joint analysis of real and simulated data revealed a component in the metabolite mode mainly modeling Ins and Glc (see $b_{2}$ in Appendix A for a comparison of models from males and females) due to the existence of such a pattern in simulated data (see $b_{2}$ in Appendix A) and that was the component revealing a BMI-related group difference. As the Ins/Glc-centric component was more tightly associated with the BMI-associated variables compared to Lac, Ala, BhB modeled in that component using the CP model of males, having a cleaner component focusing on Ins/Glc through the joint analysis improved the correlations with metavariables for males. However, in females, that component was already dominated by Ins/Glc in the CP model. Therefore, the joint analysis did not change the subject scores for females much, and correlations stayed almost the same.

When jointly analyzing real and simulated data, we gave equal importance to each data set. Determining the optimal weights in coupled factorizations remains an open research question [56,57,58]. Here, we assessed the sensitivity of the joint analysis to different weighting schemes. In other words, we jointly analyzed real and simulated data using an ACMTF model considering different α values as follows:
(5)$$\begin{array}{c} \min_{\lambda ,\sigma ,A,B,C,D,E}\alpha ∥\;X-〚\lambda ;A,B,C〛{∥}^{2}+\left(1-\alpha \right)∥\;Y-〚\sigma ;D,B,E〛{∥}^{2}+\beta {∥\;\lambda \;∥}_{1}+\beta {∥\;\sigma \;∥}_{1}\\ s.t.\;\;∥\;a_{r}\;∥=∥\;b_{r}\;∥=∥\;c_{r}\;∥=∥\;d_{r}\;∥=∥\;e_{r}\;∥=1,\mathrm{for}\;r=1,\cdots ,R \end{array}$$

Figure 8a shows that correlations with the metavariables increased as we incorporated the simulated data in the analysis. We observed that unless we were close to the extremes (i.e., α=1 which corresponded to modeling only the real data or α=0 which corresponded to modeling only the simulated data), the ACMTF model was not very sensitive to the weight selection (based on the weights considered here). From α=0.8 to α=0.1, correlations with metavariables showed minor increase. Figure 8b shows the similarity of all factors in terms of factor match scores, i.e., FMS between the factors of an ACMTF model using equal weights (i.e., α=0.5) and a different α value. We observed that models with α varying from 0.6 to 0.1 extracted similar patterns, with FMS values over 0.95. While these results support assigning equal weights to real and simulated data in the absence of prior information, learning the weights from data, for instance, by considering the noise level of each data set as in [56,57], is left as future work.

Potential discrepancies between real and simulated data sets may arise due to various reasons. For instance, there may be errors stemming from the following: (i) Model assumptions and structure: While the simulated data used in this study were constructed based on biochemical knowledge and were validated on independent data sets, the model may have omitted certain biochemical interactions, regulatory mechanisms, or external influences. Such omissions could lead to systematic deviations when the model was compared with real data. (ii) Parameters: Kinetic parameters in the simulations are often estimated or derived from the literature, introducing variability and potential inaccuracies, especially when applied to different populations or conditions. Additionally, unmodeled influences, such as environmental factors or individual-specific variations not included in the model, can further contribute to discrepancies. Therefore, it is crucial to investigate methods for detecting such conflicting information, necessitating robust diagnostic tools for detecting shared and unshared factors across data sets [45,59]. Revealing shared and unshared factors could potentially uncover new mechanisms or identify erroneous information in computational models. Consequently, such advancements would not only enhance the analysis of real data but also facilitate the improved understanding of deviations in computational models from reality [60].

As future work, we also plan to focus on different settings in simulations and study how they affect the performance. In particular, we will consider different numbers of virtual subjects to account for stronger/weaker prior information, and different levels of individual variation in the simulations. We expect that these settings will play a role in both pattern discovery and model selection using the replicability test. Furthermore, as in our experiments, simulated and real metabolomics data sets often have partially overlapping sets of metabolites. In the experiments, we focused only on the matching metabolites. We plan to consider different types of coupling between real and simulated data sets to incorporate all metabolite measurements. Recent advances in coupled tensor factorizations enable the joint analysis of data sets with such coupling relations [58].

## 5. Conclusions

Longitudinal metabolomics measurements collected over time hold the promise to improve our understanding of the metabolism, reveal early signs of diseases and facilitate precision health. Recent technological advancements have facilitated the collection of such time-resolved metabolomics measurements [61]. However, the analysis of such data sets has many challenges including noisy, missing measurements and small sample size.

In this paper, we introduced a novel data analysis approach for longitudinal metabolomics data by incorporating mechanistic models based on prior biological knowledge in order to guide the analysis of noisy real data with clean prior information. The proposed approach relied on coupled tensor factorizations, which jointly analyzed real measurements and simulated data generated by a mechanistic model in order to capture interpretable patterns and facilitate knowledge discovery from complex data. Using extensive experiments on real time-resolved metabolomics measurements and simulated data generated using a human WBM metabolic model, we demonstrated that the proposed joint analysis approach had better performance in terms of pattern discovery compared to the analysis of only real data.

While we observed promising performance in our experiments, the proposed joint data analysis approach raised further research questions such as how to detect wrong prior information, how to weigh simulated and real data sets, and how to jointly analyze partially coupled real and simulated data sets. Furthermore, the proposed approach is not limited to longitudinal metabolomics data analysis. It can also be used to guide the analysis of other types of data, e.g., neuroimaging signals or microbiome data, by mechanistic models.

## Figures and Tables

**Figure 1 metabolites-15-00002-f001:**
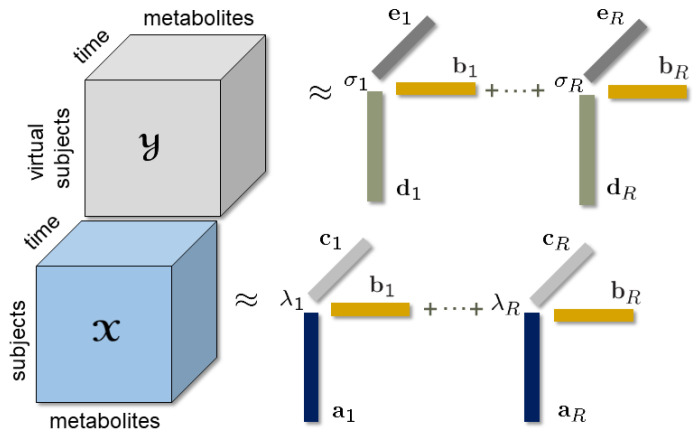
An *R*-component coupled tensor factorization jointly analyzing third-order tensors X (*subjects* by *metabolites* by *time*) and Y (*virtual subjects* by *metabolites* by *time*) coupled in the *metabolites* mode.

**Figure 2 metabolites-15-00002-f002:**
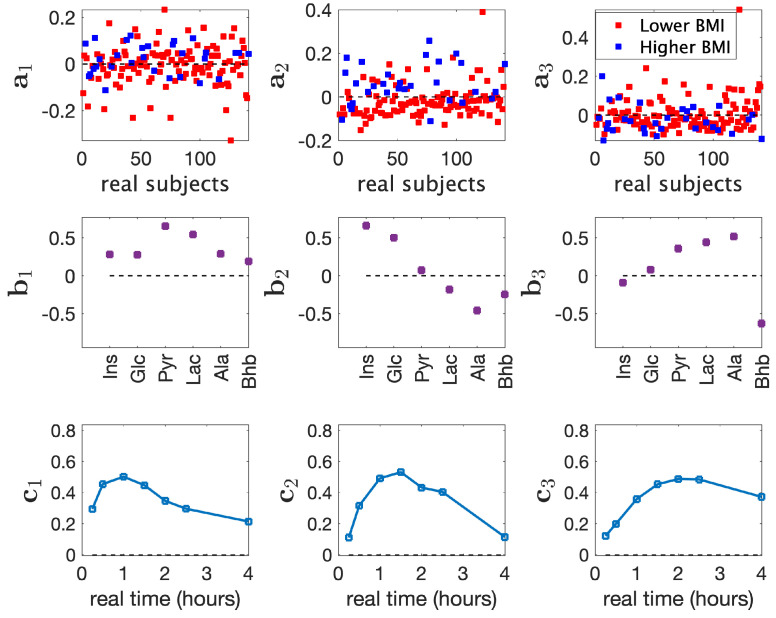
Factors of the 3-component CP model of T0-corrected data from males. $\langle a_{r},b_{r},c_{r}\rangle$, r=1,2,3, are the components in the *subjects*, *metabolites*, and *time* modes.

**Figure 3 metabolites-15-00002-f003:**
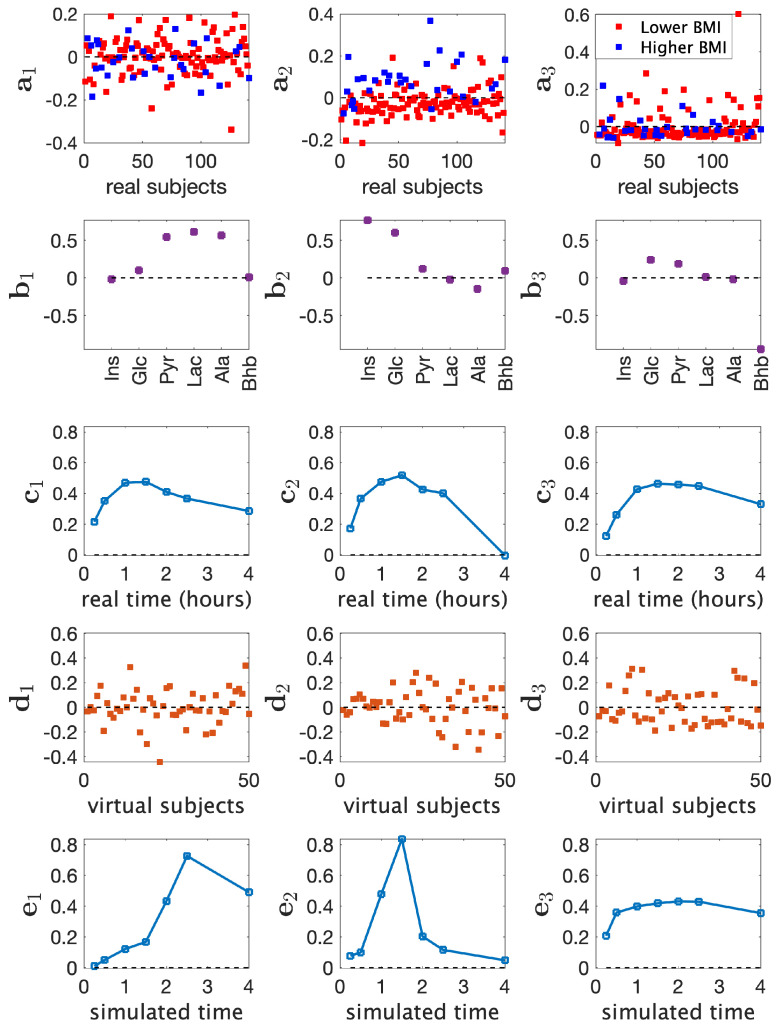
Factors of the 3-component ACMTF model of T0-corrected real data (from males) and simulated data. $\langle a_{r},b_{r},c_{r},d_{r},e_{r}\rangle$, r=1,2,3, are the components in the *subjects* (real), *metabolites* (coupled mode), *time* (real), *subjects* (virtual), and *time* (virtual) modes.

**Figure 4 metabolites-15-00002-f004:**
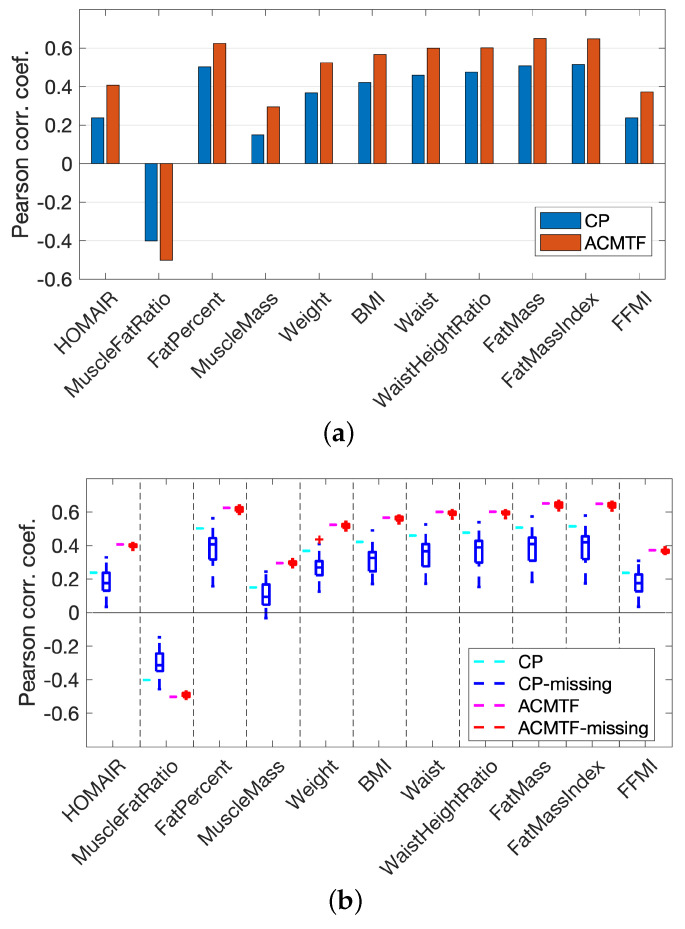
Correlations between subject scores and metavariables for the factor showing BMI-related group difference in CP and ACMTF models using (**a**) the real T0-corrected data X from males. Here, values correspond to the correlations between $a_{2}$ in Figure 2 and metavariables for CP, and between $a_{2}$ in Figure 3 and metavariables for ACMTF. (**b**) Incomplete real T0-corrected data from males, where 10% of the entries in X were removed. Thirty-two randomly incomplete data sets were considered. Correlations achieved using CP and ACMTF models of the original real data were included again in (**b**) for easier comparison. Metavariables corresponded to HOMAIR: Homeostatic model assessment for Insulin Resistance; MuscleFatRatio: muscle to fat ratio; FatPercent: body fat percentage; MuscleMass: amount of muscle in the body (kg); Weight: weight (kg); BMI: body mass index; Waist: waist circumference (cm); WaistHeightRatio: waist measurement divided by height (cm); FatMass: amount of body fat (kg); FatMassIndex: FatMass divided by ${\mathrm{height}}^{2}$; FFMI: fat-free mass index.

**Figure 5 metabolites-15-00002-f005:**
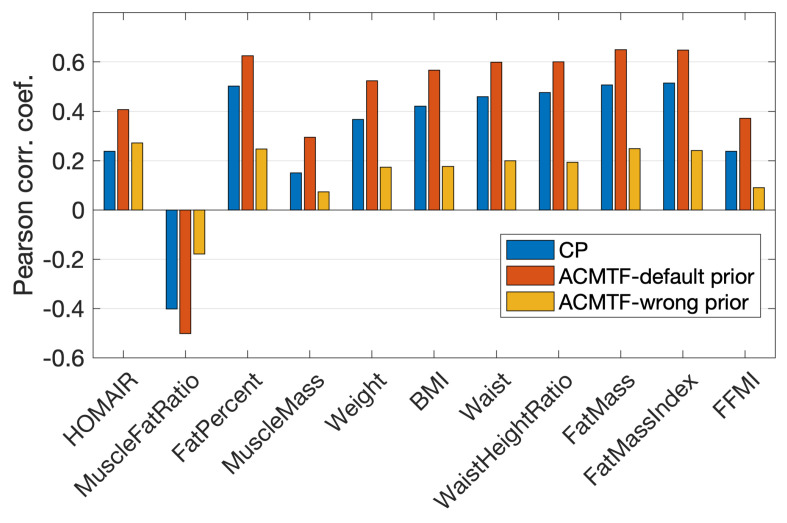
Correlations between the subject scores and metavariables for the factor showing BMI-related group difference, captured using the CP model of T0-corrected real data, ACMTF model of T0-corrected real data and the default simulated data, and the ACMTF model of T0-corrected real data and simulated data with conflicting information.

**Figure 6 metabolites-15-00002-f006:**
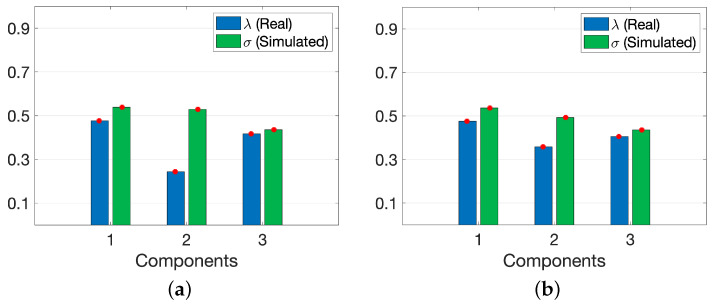
Weights of the components in ACMTF models of (**a**) T0-corrected real data and wrong simulated data, (**b**) T0-corrected real data and default simulated data.

**Figure 7 metabolites-15-00002-f007:**
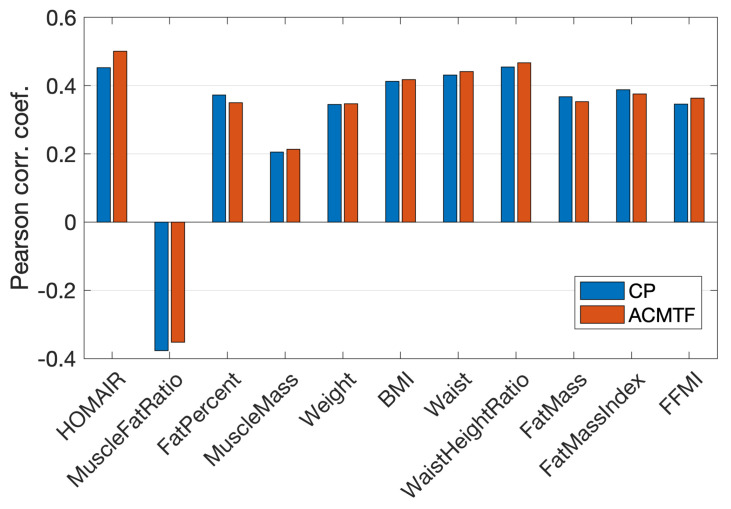
Females. Correlations between metavariables and the subject scores (for the component that revealed a statistically significant group difference in terms of BMI) using a 3-component CP model of real data and a 3-component ACMTF model of real and simulated data.

**Figure 8 metabolites-15-00002-f008:**
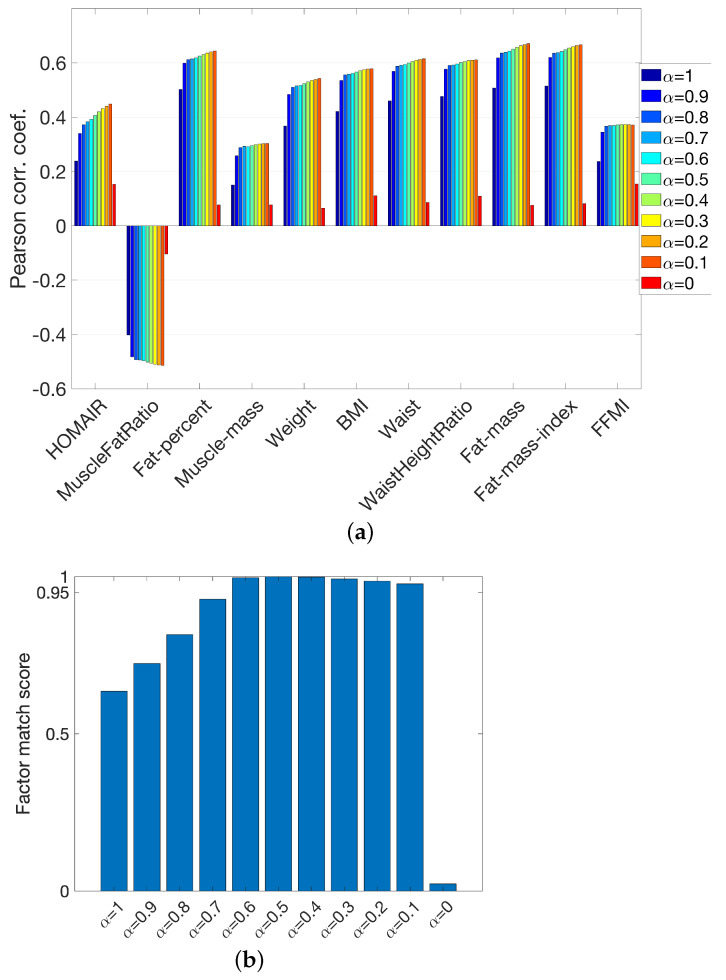
Sensitivity analysis of ACMTF models (for males) to different weighting schemes. (**a**) Correlations between the subject scores and metavariables for the factor that gave the strongest correlations, (**b**) FMS between factors extracted by an ACMTF model using different weights and those obtained with equal weights (i.e., α=0.5).

## Data Availability

The simulated data are available in the GitHub repository https://github.com/Lu-source/project-of-challenge-test-data/ (accessed on 19 December 2024). The real data (hormone and NMR measurements of plasma samples) cannot be shared publicly due to European and national GDPR. The data will be shared on reasonable request to Morten A. Rasmussen (morten.arendt@dbac.dk). The code for the joint analysis of real and simulated metabolomics data sets was released as a GitHub repository https://github.com/Lu-source/ACMTF_Real_Simulated (accessed on 19 December 2024).

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
