# Peer review of "Longitudinal Metabolomics Data Analysis Informed by Mechanistic Models"

_metabolites, 2024, doi:10.3390/metabo15010002_

Round 1
Reviewer 1 Report
Comments and Suggestions for Authors
The author presented well but comments are listed below should be address carefully,
1. Strengthen the introductory section by latest research problem, the significance of integrating simulated and real metabolomics data.
2. Emphasize on the recent literature related to this research. Tensor factorization approach, the model structure, and how real and simulated data sets are integrated in the methodology section.
3. How the metabolic components align with known physiological processes related to BMI?
4. Why certain metabolites are more prominent in males compared to females?
5. Discuss the implications of this work with recent studies and whether sex-specific models improve the result analysis or not?
6. Emphasize how different weighting schemes could affect results, especially in the context of prior knowledge. If possible, include a sensitivity analysis or discussion of how extreme cases influence outcomes.
7. Clarify the methodology used to handle missing data. Did the missing entries in the real data pose challenges during analysis? How were they addressed within the tensor factorization framework? More detail on this process could strengthen the paper.
8. Include more specific performance metrics, such as correlation coefficients, variance explained, or any statistical comparisons between the models. This will help quantify the improvements made by the joint analysis approach.
9. A more detailed discussion of the types of errors that may occur in the simulations and the potential for detecting and correcting these errors would be valuable.
10 Ensure that key figures are included in the main text or clearly referenced with brief explanations in the discussion.
Comments on the Quality of English LanguageEnglish check have to check thoroughly and avoid typos errors
Reviewer 2 Report
Comments and Suggestions for Authors
Dear Authors, dear Editor,
Draft “Longitudinal metabolomics data analysis informed by mechanistic models - metabolites-3332549-peer-review-v1” refers on a data elaboration metabolomics strategy that aims at dealing with data arrays that include missing data in small-size sample groups (“noisy” data) and time-arrays, “longitudinal measurements“, of the same subject. The Authors’ strategy is to fill the gaps by using calculated data that derive from a whole-body model and uses as input the existing data of the subject (is it?). The Authors apply their strategy to a metabolomic data array obtained by analysing with NMR plasma from subjects of an asthma cohort tested with a specific meal and from whom blood was obtained prior and at 8 time points (15, 30, 60, 90, 86 120, 150 and 240 minutes after the meal intake) during and after the meal. Some hormone measurements were obtained from the same blood samples. For each subject, also personal characteristics (sex, BMI, biometrics) were obtained.
This draft article points at methodology rather than at results, thus its description of data processing is understood by data management specialists, rather than by general readers. Who are interested in the outcomes. One detail that may mislead the common readet is:
“… six features are the common blood metabolites in the WBM 106 model, including insulin (Ins), glucose (Glc), pyruvate (Pyr), lactate (Lac), alanine (Ala) and 107 β-hydroxybutyrate (Bhb).” and “The tensor is of size 113 141 subjects × 6 metabolites × 7 time samples for males, and 152 subjects × 6 features × 7 114 time samples for females.” Explain the difference between “metabolites” (for males) and “features” (for females), or be homogeneous in terminology.
For the rest, I humbly understand that the contents can be appreciated by experts.
I leave the opinion on this draft to the Editor.
Kind regards
